# Clinical Impact of *Aspergillus fumigatus* in Children with Cystic Fibrosis

**DOI:** 10.3390/microorganisms10040739

**Published:** 2022-03-29

**Authors:** Valentina Fainardi, Chiara Sodini, Michela Deolmi, Andrea Ciuni, Kaltra Skenderaj, Maria Bice Stabile, Cosimo Neglia, Elena Mariotti Zani, Cinzia Spaggiari, Nicola Sverzellati, Susanna Esposito, Giovanna Pisi

**Affiliations:** 1Cystic Fibrosis Unit, Pediatric Clinic, Department of Medicine and Surgery, University of Parma, 43126 Parma, Italy; chiara.sodini@gmail.com (C.S.); michela.deolmi@studenti.unipr.it (M.D.); kaltra.skenderaj@gmail.com (K.S.); mariabice.stabile@studenti.unipr.it (M.B.S.); cosimo.neglia@unipr.it (C.N.); elena.mariottizani@unipr.it (E.M.Z.); spaggiaric@ao.pr.it (C.S.); susannamariaroberta.esposito@unipr.it (S.E.); gpisi@ao.pr.it (G.P.); 2Scienze Radiologiche, Department of Medicine and Surgery, University of Parma, 43126 Parma, Italy; aciuni@ao.pr.it (A.C.); nicola.sverzellati@unipr.it (N.S.)

**Keywords:** cystic fibrosis, Aspergillus, ABPA, lung function, chest CT, fungal infection

## Abstract

Background. The clinical relevance of Aspergillus fumigatus (Af) in cystic fibrosis (CF) is controversial. The aims of the study were to assess the prevalence of Af disease in our cohort of CF patients and evaluate whether allergic bronchopulmonary aspergillosis (ABPA) and sensitization to Af affected lung function, body mass index (BMI) and exacerbations. Methods. Clinical data and lung function of CF patients aged 6–18 years followed at the CF Centre of Parma (Italy) were recorded. Patients were classified as: patients with no signs of Af, patients sensitized or colonized by Af, patients with ABPA or patients with Aspergillus bronchitis (Ab). Results. Of 38 CF patients (14.2 years (6.2–18.8) M 23), 8 (21%) showed Af sensitization, 7 (18.4%) showed ABPA, 1 (2.6%) showed Af colonization and 1 (2.6%) showed Ab. Compared to non-ABPA, patients with ABPA had lower BMI (15.9 ± 1.6 vs. 19.7 ± 3.4, *p* < 0.005), lower lung function (FEV_1_ 61.5 ± 25.9% vs. 92.3 ± 19.3%, *p* < 0.001) and more exacerbations/year (4.43 ± 2.44 vs. 1.74 ± 2.33, *p* < 0.005). Patients with Af sensitization showed more exacerbations/year than non-Af patients (3.5 ± 3.2 vs. 0.9 ± 1.2, *p* < 0.005). ABPA and sensitized patients had more abnormalities on chest CT scans. Conclusion. This study showed the relevant clinical impact of ABPA and Af sensitization in terms of exacerbations and lung structural damage.

## 1. Introduction

Cystic fibrosis (CF) is a multisystem disease caused by mutation of the gene encoding for the cystic fibrosis transmembrane conductance regulator (CFTR) protein [1]. Defects in the CFTR protein result in abnormal composition of the epithelial lining fluid, viscous sputum and dysfunction of the mucociliary clearance, resulting in chronic bronchopulmonary infections [2].

Unlike bacterial infections, the impact of fungal infection on CF progression and lung destruction is still unclear. In CF patients, lung colonization by fungi is frequently observed [3]. Aspergillus fumigatus (Af) is a ubiquitous saprophytic fungus found in water, soils, decayed organic matter and indoor environments. Af spores are inhaled daily, usually with no consequences, but in people with CF, the abnormal mucus might promote the trapping of Af spores within the airways, contributing to Af colonization of the lung. Depending on fungal virulence factors and host-related factors such as immune status and pulmonary structure, the inhalation of Af can lead to a spectrum of distinct clinical entities, such as chronic colonization, Af hypersensitization, Aspergillus bronchitis or allergic bronchopulmonary aspergillosis (ABPA) [4,5,6].

The prevalence of ABPA is estimated to range from 2% to 25%, depending on diagnostic criteria and geographical area [7,8,9,10,11,12,13,14]. In contrast, sensitization to Af and sputum culture positive for Af are very common findings in CF, occurring in up to 50% of adult patients [11,14,15,16]. In children, data are scant, and the incidence of Aspergillus fumigatus in the airways may be underestimated [17]. A recent study on bronchoalveolar lavage (BAL) showed that Af might be present in up to 28% of patients from 3 years of age onwards [18], suggesting that this fungus might also be very common at an early age. Numerous studies in CF patients have demonstrated an association between ABPA and poorer clinical outcomes [18,19,20], and some evidence shows that chronic Af infection and Af sensitization are also associated with structural lung disease, reduced lung function and pulmonary exacerbations [12,16,19,21,22]. However, the impact of fungal infection on CF progression remains unclear, and clinical guidelines on the management of the different forms of Af disease are lacking.

The aim of this study was to assess the prevalence of Aspergillus disease in a cohort of children with CF and its influence on lung function and structure, body mass index (BMI) and exacerbation rate.

## 2. Methods

### 2.1. Study Population

CF patients aged 6–18 years regularly seen at Parma Cystic Fibrosis Regional Centre (Italy) during the year 2020 were identified by medical records, and retrospective data were collected for the time period 2018–2020. Exclusion criteria were coexistent malignancies and lung transplantation. All had a diagnosis of CF confirmed by a positive sweat test with chloride concentration > 60 mmol/L and DNA analysis of a CF-specific genotype. Annual blood tests including total and specific IgE antibodies against Af are usually obtained for all patients followed at the center. Bacterial and fungal cultures are performed routinely on sputum or throat swabs 4 times/year as part of clinical care; airway samples are processed in the local microbiology laboratory following standard CF culture procedures. Spirometry is performed 4 times/year on a regular basis.

The study was approved by the Ethical Committee of Emilia-Romagna Area Vasta Nord (Prot. N. 24041, Date: 6 April 2021). Before inclusion, the parents signed written informed consent, and the patients provided written assent.

### 2.2. Data Collection and Definitions

Anthropometric (gender, age, weight, height and BMI) and clinical (CFTR gene mutation, pancreatic function, number of pulmonary exacerbations per year and pharmacological treatment) data, laboratory findings (sputum culture, skin test for *Aspergillus fumigatus* (Af), total serum immunoglobulin (Ig) E, serum-specific IgE for Af, eosinophil count and specific serum precipitins for Af), lung function (FEV_1_ measured by spirometry) and high-resolution computed tomography (HRCT) results (where available) were collected over the period 2018–2020.

According to the ISHAM Working Group [23,24], ABPA was diagnosed when the following criteria were met: (i) positive skin test for Af or high levels of serum-specific IgE for Af AND high total serum IgE (>1.000 IU/mL); (ii) plus at least two of these criteria: high levels of Aspergillus-specific precipitating antibodies, chest X-ray suggestive for ABPA or total eosinophil count higher than 500 cells/mL.

ABPA stage was defined as [25]:Stage 1 (acute phase): typical findings are present (serum-specific IgE for Aspergillus, Aspergillus-specific precipitating antibody, radiological abnormalities, peripheral blood eosinophilia).Stage 2 (remission phase): asymptomatic patient, no new radiological infiltrates and no increase in total serum IgE for at least 6 months.Stage 3 (exacerbation): new pulmonary infiltrates with eosinophilia and doubled IgE levels with respect to remission phase.Stage 4: steroid-dependent disease.Stage 5 (pulmonary fibrosis): chest X-ray or CT scan shows irreversible fibrosis and chronic cavitation.

Sensitization to Af was defined by the presence of a positive prick test for Af antigen or raised IgE against Af, while colonization was considered to be present when patients had at least two positive sputum cultures for Af in the previous 12 months [26]. Chronic *Aspergillus* bronchitis was diagnosed when *Aspergillus*-specific precipitating antibodies were detected together with clinical deterioration but no other criteria of ABPA were present [27]. Exacerbation was defined as the deterioration of clinical condition and/or increase in productive cough requiring a course of antibiotics. The presence of *Stenotrophomonas maltophilia*, *Achromobacter xylosoxidans* or *Mycobacterium abscessus* in the sputum culture classified the patient as colonized by multidrug-resistant bacteria, as reported elsewhere [28].

Global Lung Function Initiative (GLI)-2012 reference equations were used to determine the percent of predicted forced expiratory volume in the 1st second (FEV_1_) based on age, sex, height and ethnicity [29].

### 2.3. HRCT Evaluation and Scoring System

Patients were scanned with a 128-slice Somatom Definition Flash scanner (Siemens Medical Solutions, Forchheim, Germany). Images were visually scored using a window setting (−1550 W, −600 L) by a radiologist blinded to clinical information using the Bhalla scoring system [30]. The total score was the sum of the following sub-scores: severity of bronchiectasis, peribronchial thickening, extent of bronchiectasis, extent of mucus plug, sacculation or abscesses, bronchial generation with bronchiectasis and/or mucus plug, distribution and total number of bullae, extent of emphysema and extent of collapse and/or consolidation. The Bhalla score was calculated by scoring each of the 9 categories. The total points were then subtracted from 25 to obtain the Bhalla score. The Bhalla score ranges between 0 and 25, where a lower score indicates more severe radiological lung abnormalities.

### 2.4. Statistical Analysis

For the descriptive analysis of the continuous variables, the mean and standard deviation (SD) were calculated, and frequencies were reported for dichotomous and qualitative variables. Non-parametric Wilcoxon rank-sum test was performed to compare the means of continuous variables between two groups. For categorical and dichotomous variables, differences between groups were tested by using the chi-square test and Fisher exact test for frequencies less than or equal to 5. To describe FEV_1_ decline over the years, for each patient, the mean FEV_1_ value per year was obtained, and then the difference between 2018 and 2020 was calculated. To describe the effect of ABPA and quantitative variables such as CT score and BMI on FEV_1_ percent predicted, a linear mixed-effects regression model with a random intercept was applied. Patients with Af colonization and Aspergillus bronchitis were not considered as single groups but excluded from the non-Af group for the analysis. Data were analyzed using SPSS version 26. A two-tailed *p*-value of <0.05 was considered statistically significant. Data analysis was performed using SPSS version 26 and STATA Statistical Software (Release 11 College Station, TX, USA).

## 3. Results

One hundred and eighty-one CF patients (median age 27 years, range 0–66, 93 M) were regularly followed-up at the center at the time of data collection (1 December 2020). Forty patients aged 6–18 years were included in the analysis. One patient was excluded because she suffered from Ewing sarcoma, and one was excluded because he had a history of lung transplantation. Figure 1 describes the study population.

### 3.1. Prevalence of Af-Related Disease

At the time of data collection, 7 patients (18%) fulfilled all criteria for ABPA, 8 (21%) were sensitized to Af, 1 (3%) was colonized by Af, and 1 (3%) showed Aspergillus bronchitis; 21 (55%) subjects had no signs of Af infection, sensitization or colonization. No difference was noted in sex or age among the groups.

Characteristics and laboratory data of the four groups at the time of data collection are shown in Table 1 and Table 2, respectively. Four patients with no signs of Af did not have longitudinal data of lung function (two never performed spirometry, and two had only one measurement).

Compared to the rest of the cohort, patients with ABPA showed lower BMI (15.9 ± 1.6 kg/m^2^ vs. 19.7 ± 3.4 kg/m^2^, *p* < 0.005) and lower lung function (FEV_1_ 61.5 ± 25.9% predicted vs. 92.2 ± 19.3% predicted, *p* < 0.001) (Figure 2), were more likely to be pancreatic insufficient (*p* < 0.05) and were more likely to use long-term inhaled antibiotics (tobramycin, colistin or levofloxacin) or receive long-term treatment with oral azithromycin and inhaled steroids (*p* < 0.005).

Over the previous 12 months, patients with ABPA experienced the highest number of exacerbations (4.43 ± 2.44 vs. 1.74 ± 2.33, *p* < 0.005). Compared to patients with no signs of Af, a higher exacerbation rate was also reported in sensitized patients (3.5 ± 3.25 vs. 0.9 ± 1.26, *p* < 0.005).

For the whole study cohort, FEV_1_ decreased by approximately 7% over three years (median −6.9%, range −48.3%–35%) (Figure 3). The greatest decline in lung function was observed in patients with ABPA (−27 ± 19.1% vs. −2 ± 14%, *p* < 0.001).

At the time of data collection, patients showed different stages of ABPA: six were in stage 3 (“exacerbation”), and 1 was in stage 2 (“remission phase”).

### 3.2. ABPA Effect on Lung Function

In a linear regression model, ABPA was associated with a decrease of 27.5% in the FEV_1_ value. Although not significant, an increase of 1 point in BMI was associated with an increase of 1.3% in FEV. Parameter estimates of the full linear mixed-effects regression model describing the associations of FEV_1_ percent predicted with BMI, CT score and ABPA are shown in Table 3.

### 3.3. Changes in Lung Structure

HRCT scans were available for 29 out of 38 patients (76.3%), of which six had ABPA, and eight were sensitized to Af. Compared to patients with no signs of Af, patients with ABPA and patients sensitized to Af showed more abnormalities in lung structure (Bhalla scores: ABPA 14 ± 3.6, Af sensitization 17.1 ± 5.8, non-Af patients 22.2 ± 2.8; *p* < 0.005). Figure 4 shows typical radiological features found in ABPA patients.

## 4. Discussion

Our study showed that in CF children, the prevalence of ABPA and sensitization to Af are quite high, with a negative impact on BMI, pulmonary function and structural lung disease.

The prevalence of ABPA in our population was very similar to the prevalence reported by Kaditis et al. in a recent large epidemiological study including 3550 patients aged 6–17 years (18% in our cohort vs. 18.2%) [13]. Similarly, the prevalence of sensitization to Af in our cohort was comparable to the prevalence described by Gothe et al. in a cohort of 387 children and young adults with CF (21% vs. 19%) [31]. However, the prevalence of sensitization differs widely among studies, as demonstrated by a meta-analysis of 41 studies including children and adults (n = 3362 patients), where the prevalence varied from 20% to 65%, with a pooled prevalence of 39.1% [14]. The reasons for this heterogeneity might be the different ages of the patients, the geographical area or the different methods used to detect sensitization (skin tests have higher sensitivity compared to specific IgE levels in determining fungal sensitization) [14].

In our cohort, children with ABPA had, on average, a 30% lower FEV_1_ value than those without ABPA. In a cohort of 12,447 pediatric and adult CF patients participating in the European Epidemiologic Registry of Cystic Fibrosis (ECFSPR), Mastella et al. showed an FEV_1_ reduction of about 10% in those with ABPA [12]. A similar difference in the mean FEV_1_ value (approximately 9%) between patients with and without ABPA was found in the more recent report by Kaditis et al. on 3350 patients aged 6–17 years. However, this difference in FEV_1_ decreased to 1.47 percentage points when considering other co-existing adverse prognostic factors, such as low BMI, severity of genotype, female gender, CF-related diabetes or chronic P. aeruginosa infection [13]. The greater difference in lung function found in this study compared to other studies in the literature might be due to a longer time spent with ABPA, which could have impacted disease severity. Furthermore, at the time of data collection, six out of seven patients with ABPA showed an exacerbation of ABPA disease, suggesting a more severe clinical picture and lower lung function values.

Compared to the rest of the cohort, in patients with ABPA, BMI was significantly lower. Low BMI has been considered a risk factor for ABPA; however, ABPA disease might have contributed to weight loss, and the true relationship between ABPA and BMI must be further studied [32].

Compared to children without signs of Af, we demonstrated a greater FEV_1_ decline in ABPA patients over a 3-year period (−27 ± 19.1% vs. −2 ± 14%), in accordance with previous authors [14,21,33]. However, the two largest longitudinal studies reported no effect of ABPA on the annual decline over a period of 3 and 5 years, respectively [12,13]. Harun et al. confirmed this finding in a cross-sectional analysis in 5-year-old subjects from the Australasian Cystic Fibrosis Bronchoalveolar Lavage Study. The study suggested that a positive Aspergillus culture had no significant impact on predicted lung function decline between ages 5 years and 14 years [34]. Conversely, a longitudinal study on lung function in children with CF identified not only ABPA but also Aspergillus colonization as an independent risk factor for a greater increase in the lung clearance index over time [35]. In another study, AlShakirchi et al. showed that Af colonization was associated with lower lung function and accelerated lung function decline, while its eradication resulted in better lung function [36]. These contrasting findings about the influence of ABPA on lung function decline can have different explanations. First, the results reported by the first two longitudinal studies were calculated after adjustment for other adverse prognostic factors, and the size of the cohorts was much greater than ours. In addition, the onset and the stage of the disease were uncertain, and patients might have been in the remission phase of the disease, when lung function deficit may have been lower. In our cohort, almost all patients showed exacerbation of ABPA at the time of data collection, and the lower FEV_1_ at baseline due to the active phase of the disease may have influenced the results and the comparison with the other subjects.

In support of this observation, we observed that children with a diagnosis of ABPA suffered from more pulmonary exacerbations in the previous 12 months and showed more lung damage compared to subjects never affected by Aspergillus disease. Almost all patients with ABPA (6/7) were colonized by Pseudomonas or multiresistant bacteria, and these microorganisms may have also contributed to the development of respiratory exacerbations and to the progression of lung disease. The concomitant presence of Af and other germs has been demonstrated as a risk factor for ABPA and disease severity [32,37]. The Irish CF registry reported that coinfection with P. aeruginosa and Af was associated with a 165% increase in hospital admissions and a 112% increase in the number of respiratory exacerbations compared to patients with negative cultures for both pathogens [38]. ABPA has been associated with significant structural lung damage in patients with CF, particularly air trapping [39], which, in young children with CF, correlates with small airway structural lung disease. We could not compare multiple HRCT scans over time to assess the correlation between lung function decline and lung structure deterioration, but a recent study demonstrated that children with ABPA had significantly greater development of structural lung disease over a period of two years despite a stable FEV_1_ [39].

Despite normal lung function, patients sensitized to Af also had a higher number of exacerbations compared to children with no signs (colonization, infection or sensitization) of Af. Interestingly, these patients revealed more structural abnormalities on chest HRCT scans in terms of trapped air and mucus plugging. Both the higher exacerbation rate and the anomalies seen on HRCT suggest that Af may have a damaging impact on the lung and the clinical condition despite a normal spirometry. An increased number of exacerbations [40] and early lung abnormalities on HRCT were reported in both colonized and sensitized patients [18,34,41]. Two recent studies based on BAL cultures of children with CF found that Af infections were associated with air trapping [18,34] and increased progression of structural lung disease within 1 year after the infection [9]. Similar to the observations in our study, children with these abnormalities on HRCT scans had normal lung function [34]. Since chest HRCT scores are more sensitive than spirometry at detecting early and progressive CF lung disease in children [39,42], we may speculate that in these patients, lung function tests, such as multiple breath washout or impulse oscillometry, could have been more sensitive in the detection of small airway impairment. However, studies have failed to demonstrate a correlation between HRCT scores and lung function values [39,43,44]. Taken together, these results may suggest that anomalies seen on HRCT scans are related to the Aspergillus-related disease and not to the CF disease and that early eradication of Af from the airways could avoid some long-term negative effects on lung structure. Some preliminary studies reported interesting results on the use of magnetic resonance imaging (MRI) to diagnose ABPA from the specific finding of inverted impaction mucus [45,46].

Several risk factors have been proposed for the development of Af infection and colonization in CF patients: female gender [13], pancreatic insufficiency [37], low BMI [32], CF-related diabetes, chronic administration of macrolides, inhaled antibiotics or inhaled steroids [47,48] and Pseudomonas aeruginosa or Stenotrophomonas maltophilia infection [32,47,48,49]. With the limitations of a small cohort, in our study, ABPA prevalence was higher in males, and almost all patients of the group had pancreatic insufficiency, low BMI and low lung function; presented coinfection of Aspergillus with P. aeruginosa or multiresistant bacteria; and used long-term azithromycin, inhaled antibiotics or inhaled steroids more frequently than other patients. Pancreatic insufficiency, low BMI and the greater use of pharmacological treatments are all indicators of disease severity that may have contributed to the development of ABPA. Furthermore, the presence of P. aeruginosa and multiresistant bacteria in almost all patients with ABPA may suggest that the high use of antibiotics administered in the attempt to eradicate these bacteria could have been a further predisposing factor for Af colonization and ABPA development [50]. Conversely, the contribution of ABPA disease in reducing BMI and lung function and therefore increasing the use of medications is difficult to assess. However, after controlling for baseline variables known to be associated with worse pulmonary disease, in our cohort, ABPA seems to have played a pivotal role in lung function decline and disease progression.

Our study has some limitations. It is an observational, retrospective study, performed in a small cohort in a single center. Therefore, associations between Af-related disease and clinical outcomes are difficult to make. Furthermore, we started data collection only as early as 2019, and some patients may have started presenting signs and symptoms of Af-related disease earlier. Total IgE can spontaneously decrease, and specific IgE or precipitins against Af can become negative over time in a certain percentage of cases, preventing clinicians from making the diagnosis [11].

This study could not determine whether ABPA or sensitization to Af is a direct mediator of CF lung impairment or if patients with advanced lung disease are to a greater extent more susceptible to A. fumigatus disease. However, our data confirm previous studies where patients with ABPA had impaired lung function and required more hospitalizations, suggesting a severe clinical picture [48,51]. Prospective longitudinal studies are needed to assess the risk factors for Af-related disease and to explore the interactions between Af and other microorganisms such as P. aeruginosa in the airways of CF patients.

## 5. Conclusions

This study showed that Af played a pivotal role in pediatric patients with CF and that ABPA and Af sensitization had a relevant clinical impact in terms of exacerbations and lung structural damage. These findings highlight that annual screening for total serum IgE levels and Aspergillus-specific IgE levels is essential to intercept the onset of Aspergillus sensitization and/or signs of ABPA. Early diagnosis and regular treatment of ABPA seems crucial to prevent serious and potentially irreversible lung damage. Patients colonized or chronically infected with P. aeruginosa or multiresistant microorganisms should be regularly tested for A. fumigatus sensitization. In addition, eradication of Aspergillus in the sensitized patient could lead to a better clinical outcome by preventing progression to ABPA and limiting lung damage. However, in CF patients with Af, the precise timing for initiating treatment is still controversial. Similarly, it is unclear whether individuals sensitized to Aspergillus might benefit from early eradication treatment and if it could somehow prevent the development of ABPA. Imaging techniques play a key role in the management of Aspergillus-related disease. Chest HRCT can differentiate between harmless colonization and infection, allow early identification of structural abnormalities associated with A. fumigatus infection and therefore support the decision to start pharmacological treatment. Further studies on chest MRI could improve the diagnosis of Af disease and allow close monitoring of the disease and treatment over time without exposing the patient to radiations [52].

## Figures and Tables

**Figure 1 microorganisms-10-00739-f001:**
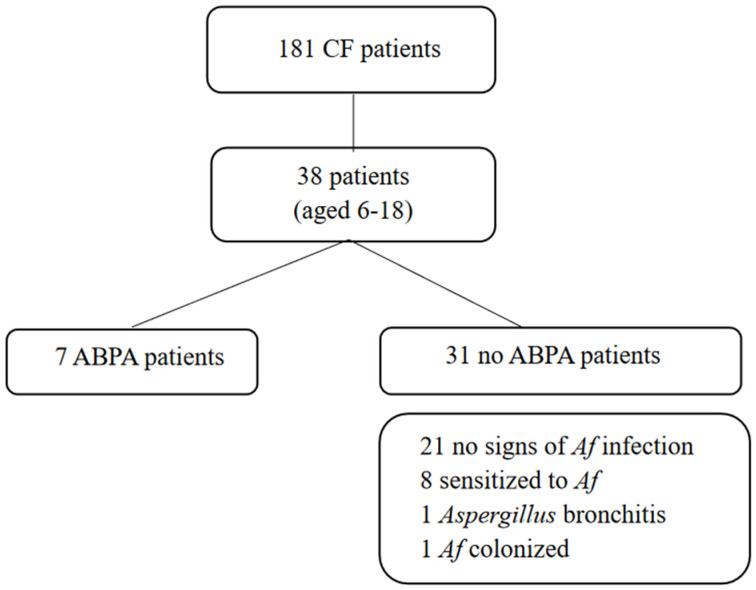
Study population.

**Figure 2 microorganisms-10-00739-f002:**
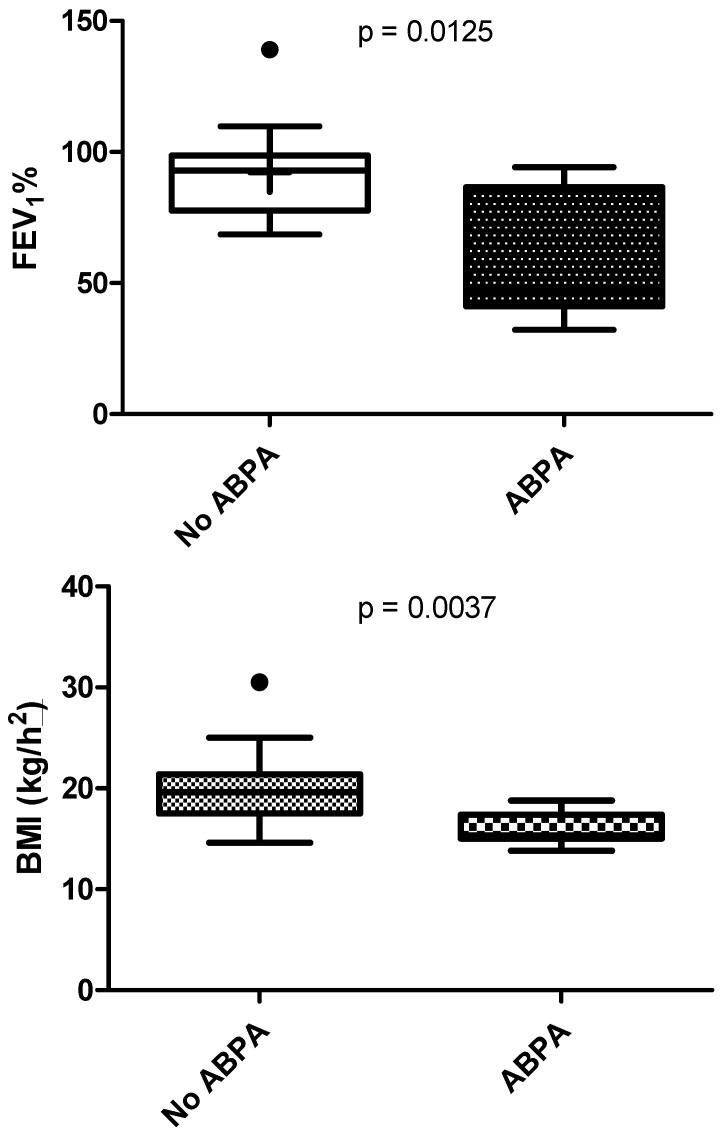
Difference in FEV_1_ and BMI between patients with and without ABPA.

**Figure 3 microorganisms-10-00739-f003:**
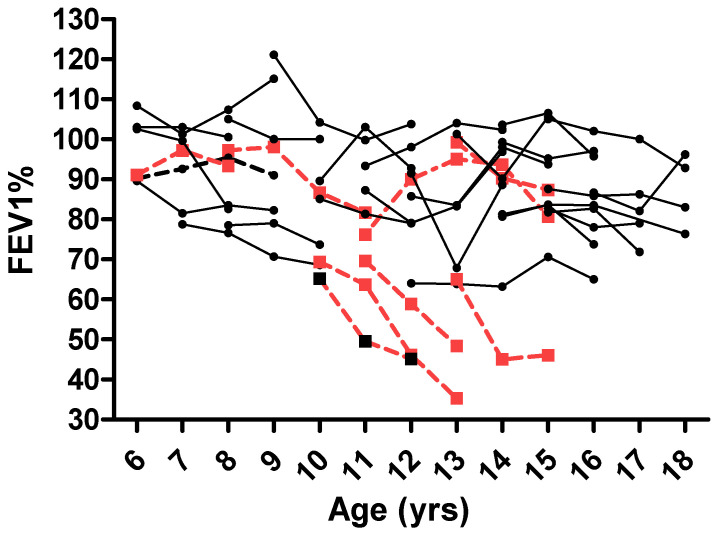
Lung function progress (FEV_1_ %) for the whole population in the study period. Dotted lines and square symbols refer to patients with ABPA.

**Figure 4 microorganisms-10-00739-f004:**
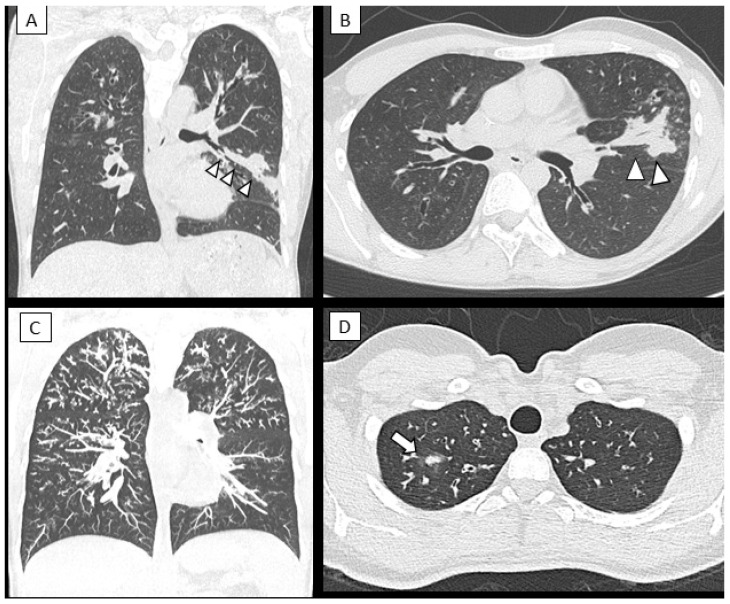
CT findings of mucus plugging: multi-planar reconstruction (MPR) in a coronal (**A**) and axial oblique plane (**B**) of a 17-year-old boy. The “finger in glove sign” (arrowheads) can be seen in the lingual region; this sign is consistent with large airway mucoid impaction, frequently seen in ABPA. Other parenchymal and airway abnormalities are seen in the apical regions of the lungs (cylindrical bronchiectasis and nodular consolidation). Coronal maximum intensity projection-MIP (**C**) and axial plane CT images (**D**) of a 16 year-old female with ABPA shows diffuse branched opacities in the upper lung regions, so called “tree in bud appearance” representing distal airways mucus plugging. Small consolidation in the apical segment of the right upper lobe is also seen (white arrow).

**Table 1 microorganisms-10-00739-t001:** Characteristics of patients at time of enrollment. All values are expressed as mean ± SD.

	No ABPA					ABPA (*n* = 7)	*p*-Value
	No Signs of Af (*n* = 21)	Af Sensitization (*n* = 8)	Af Colonization (*n* = 1)	*Aspergillus* Bronchitis (*n* = 1)	Total No ABPA (*n* = 31)		
**Age, mean ± S D**	12.9 ± 4.3	14.6 ± 2.9	17.7	16.2	13.3 ± 3.9	13.4 ± 2.5	ns
**Sex**							
Male	11 (52.4)	5 (62.5)	1 (100.0)	1 (100.0	18 (58.1)	5 (81.4)	ns
Female	10 (47.6)	3 (37.5)	-	-	13 (41.9)	2 (28.6)	
**BMI (kg/m^2^), mean ± SD**	20.36 ± 3.61	19.79 ± 2.48	19.6	17.6	19.7 ± 3.41	15.96 ± 1.68	<0.005
** Pancreatic insufficiency, *n* (%) **	5 (24)	5 (62.5)	1 (100)	1 (100)	12 (38.7)	6 (86)	<0.05
** CFTR gene mutation, *n* (%) **							<0.001
Homo-F508del	2 (9.5)	3 (37.5)	1 (100)	1 (100)	7 (22.5)	2 (28.5)	
Hetero-F508del	15 (71.5)	3 (37.5)	-	-	18 (58)	2 (28.5)	
Others	4 (19)	2 (25)	-	-	6 (19.3)	3 (43)	
**FEV_1_, % predicted**	91.3 ± 18.9	96.8 ± 22.3	71.9	95.3	92.3 ± 19.3	61.5 ± 25.9	<0.001
**FEV_1_ change over 3 years (%)**	−3.7 ± 10.2	9.14 ± 14.1	−14.9	−39	2 ± 14	−27 ± 19.1	<0.001
**Comorbidities**							
Diabetes, *n* (%)	1 (4.7)	2 (25)	0	1	4 (12.9)	0	ns
**CT score**	22.2 ± 2.8	17.1 ± 5.8	15	12	19.7 ± 5	14 ± 3.6	<0.005
**Exacerbations in the last 12 months**	0.9 ± 1.26	3.50 ± 3.25	1	6	1.74 ± 2.33	4.43 ± 2.44	<0.005
**Sputum culture**							
*Pseudomonas aeruginosa*, *n* (%)	9 (42.9)	1 (12.5)	0 (0)	1 (100)	10 (32.2)	3 (42.8)	ns
*Aspergillus*, *n* (%)	0 (0)	2 (25)	0 (0)	0 (0)	2 (6.4)	0 (0)	ns
*Pseudomonas aeruginosa* AND *Aspergillus*, *n* (%)	0 (0)	2 (25)	1 (100)	0 (0)	3 (9.6)	2 (28.5)	ns
*Multiresistant bacteria*, *n* (%)	4 (19)	2 (25)	1 (100)	1 (100)	8 (25.8)	0 (0)	ns
*Multiresistant bacteria* AND *Aspergillus*, *n* (%)	0 (0)	0 (0)	0 (0)	0 (0)	0 (0)	1 (14.2)	ns
**Treatments**							
Long-term inhaled antibiotics, *n* (%) *	1 (4.7)	2 (25)	0 (0)	1 (100)	4 (12.9)	4 (57.2)	<0.05
Oral azithromycin, *n* (%)	1 (4.7)	3 (37.5)	1 (100)	1 (100)	6 (19.3)	4 (57.2)	0.007
Nebulized hypertonic saline, *n* (%)	1 (4.7)	0 (0)	0 (0)	0 (0)	1 (3.2)	1 (14.3)	<0.001
Inhaled corticosteroids, *n* (%)	6 (28.6)	5 (62.5)	1 (100)	1 (100)	13 (41.9)	6 (85.8)	<0.05

SD—standard deviation; Af—*Aspergillus fumigatus*; BMI—body mass index; FEV_1_—forced expiratory volume in the first second; multiresistant bacteria: *Stenotrophomonas maltophilia*, *Achromobacter xylosoxidans* or *Mycobacterium abscessus*; * tobramycin, colistin or levofloxacin.

**Table 2 microorganisms-10-00739-t002:** Laboratory data of patients at time of enrollment. All values are expressed as mean ± SD.

	No Signs of Af (*n =* 21)	Af Sensitization (*n =* 8)	ABPA (*n =* 7)	Af Colonization (*n =* 1)	Aspergillus Bronchitis (*n =* 1)
**Eosinophil count**	0.24 ± 0.20	0.31 ± 0.17	0.55 ± 0.37	0.16 ± 0.04	0.24 ± 0.09
**Total IgE**	91.5 ± 101.7	214.6 ± 182.5	918.4 ± 509.3	27.83 ± 6.62	110.10 ± 56.15
**Specific IgE for Aspergillus fumigatus**	0 ± 0.01	1.10 ± 1.48	12.51 ± 6.56	<0.01	0.12 ± 0.11
**Aspergillus specific precipitating antibodies**	17.4 ± 140.8	42.4 ± 240.2	64.9 ± 340.1	44.3 ± 4.0	>100
**Af in sputum, *n* (%)**	0 (0)	4 (50)	4 (57)	1 (100)	0 (0)

SD—standard deviation; ABPA—allergic bronchopulmonary aspergillosis; Af—*Aspergillus fumigatus*.

**Table 3 microorganisms-10-00739-t003:** Parameter estimates of the full linear mixed-effects regression model describing the associations of FEV_1_ percent predicted with BMI, CT score and ABPA.

Variable	Coefficient	95% CI	*p*-Value
BMI	1.32	[−0.98–3.63]	0.25
CT score	0.38	[−1.11–1.87]	0.60
ABPA	−27.64	[−47.23–−8.05]	<0.01
Constant	57.61	[9.92–105.9]	<0.05

R-squared: 0.46. No. observations: 38.

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
