# Peer review of "Clinical Impact of Aspergillus fumigatus in Children with Cystic Fibrosis"

_microorganisms, 2022, doi:10.3390/microorganisms10040739_

Round 1

Reviewer 1 Report

This study investigated the prevalence of Af disease in children with CF and assess the impact of ABPA and Af sensitization on the outcome of these populations. Overall, the study is interesting. However, I have several concerns.

  1. This manuscript needs extenstive English editing.
  2. This study had a major limitation - small case number, which could be due to rare diseases. 
  3. Please specifically describe the inhaled antibiotics and oral macrolide.
  4. Figure 3 is difficult to read. Please marked the group with or without ABPA
  5. Please define multidrug resistant pathogens.

Author Response

We thank the reviewer for the comments. A point by point response is provided below.

  1. This manuscript needs extenstive English editing. The text has been improved.
  2. This study had a major limitation - small case number, which could be due to rare diseases. This has been discussed among the limitations of the study in the discussion.
  3. Please specifically describe the inhaled antibiotics and oral macrolide. Inhaled antibiotics have been specified in the table and in the methods section. "Oral macrolide" has been changed with "azithromycin".
  4. Figure 3 is difficult to read. Please marked the group with or without ABPA. Thanks for the comment. The figure has been changed highlighting lung function of ABPA patients.
  5. Please define multidrug resistant pathogens. Multidrug resistant pathogens have been described in the methods section.

Reviewer 2 Report

The manuscript entitled "Clinical impact of Aspergillus fumigatus in children with cystic fibrosis", by Valentina Fainardi  and colleagues, describes the study of prevalence of Aspergillus fumigatus  disease in a cohort of cystic fibrosis patients aged 6-18 years followed at the CF Centre of Parma (Italy), focusing on the allergic bronchopulmonary aspergillosis and sensitization to Aspergillus fumigatus affected lung function, body mass index (BMI) and exacerbations. This is a retrospective study with some limitations as indicated by the authors, mainly due to the limited number of patients enroled and the limited information on the time of infection by Aspergillus and the course of the infection/colonization. Nevertheless, it represents a good contribution to the knowledge on the clinical significance of Aspergillus fumigatus colonization/infections among cystic fibrosis patients, a topic tha thas remained unclear. Despite the limited number of patients and other limitations, the study is well organized, the methods used are adequate and the conclusions taken are supported by the results obtained. The manuscript is well written and organized, although there a few  minor issues that require the attentio of the authors. 

line 44: Instead of "Prevalence of ABPA is estimated to be between 2% and 25%...", I suggest to write "Prevalence of ABPA is estimated to range from  2% to 25%...".

Line 47: italicize "Apergillus fumigatus".

Table 1 Italicize "Pseudomonas", "Aspergillus".

Table 2: An entry mentioning multi-resistant bacteria is expalined in the table footnote as Stenotrophomonas maltophilia, Achromobacter xylosoxidans, Mycobacterium. Why is this group of bacteria mentioned as multiresistant? There is no mention in the text that these bacteria were chracterized as multiressistant. Clinical isolates of the species mentioned are often multiresistant, but a clearer explanation should be provided by the authors.

Page 11, 3rd paragraph, line 6: correct "sect3ional"

Page 13, line before the "5. Conclusions": italicize P. aeruginosa.

Author Response

We thank the reviewer for the comments. A point by point response is provided below.

line 44: Instead of "Prevalence of ABPA is estimated to be between 2% and 25%...", I suggest to write "Prevalence of ABPA is estimated to range from  2% to 25%...". Thank you for the comment. The sentence has been changed accordingly.

Line 47: italicize "Apergillus fumigatus".

Table 1 Italicize "Pseudomonas", "Aspergillus".

All names of pathogens have been written in italic.

Table 2: An entry mentioning multi-resistant bacteria is expalined in the table footnote as Stenotrophomonas maltophilia, Achromobacter xylosoxidans, Mycobacterium. Why is this group of bacteria mentioned as multiresistant? There is no mention in the text that these bacteria were chracterized as multiressistant. Clinical isolates of the species mentioned are often multiresistant, but a clearer explanation should be provided by the authors.

Thank you for the comment. We clarified in the methods section and we added a reference specific for CF to justify the classification as multiresistant bacteria.

Page 11, 3rd paragraph, line 6: correct "sect3ional". This typing error has been corrected.

Page 13, line before the "5. Conclusions": italicize P. aeruginosa.

Round 2

Reviewer 1 Report

The authors response well, so I have no more comment.